# A two-arm parallel-group individually randomised prison pilot study of a male remand alcohol intervention for self-efficacy enhancement: the APPRAISE study protocol

Aisha Holloway,[1] Victoria Guthrie [ID] ,[1] Gillian Waller,[2] Jamie Smith,[1,3] Joanne Boyd,[2,4] Sharon Mercado,[1] Pam Smith,[1] Rosie Stenhouse [ID] ,[1] Aziz Sheikh,[5] Richard Anthony Parker,[6] Andrew Stoddart,[7] Philip Conaglen,[8] Simon Coulton,[9] Gertraud Stadler,[3,10] Kate Hunt,[11] Jeremy Bray,[12] Jennifer Ferguson,[2] Arun Sondhi,[13] Kieran Lynch,[14] Jessica Rees,[6] Dorothy Newbury-Birch[2]

**Correspondence to**
Professor Aisha Holloway;
Aisha.Holloway@ed.ac.uk

## ABSTRACT

**Introduction** The prevalence of at-risk drinking is far higher among those in contact with the criminal justice system (73%) than the general population (35%). However, there is little evidence on the effectiveness of alcohol brief interventions (ABIs) in reducing risky drinking among those in the criminal justice system, including the prison system and, in particular, those on remand. Building on earlier work, A two-arm parallel group individually randomised Prison Pilot study of a male Remand Alcohol Intervention for Self-efficacy Enhancement (APPRAISE) is a pilot study designed to assess the feasibility and acceptability of an ABI, delivered to male prisoners on remand. The findings of APPRAISE should provide the information required to design a future definitive randomised controlled trial (RCT).

**Methods and analysis** APPRAISE will use mixed methods, with two linked phases, across two prisons in the UK, recruiting 180 adult men on remand: 90 from Scotland and 90 from England. Phase I will involve a two-arm, parallel-group, individually randomised pilot study. The pilot evaluation will provide data on the likely impact of A two-arm parallel group individually randomised Prison Pilot study of a male Remand Alcohol Intervention for Self-efficacy Enhancement (APPRAISE), which will be used to inform a future definitive multicentre RCT. Phase II will be a process evaluation assessing how the ABI has been implemented to explore the change mechanisms underpinning the ABI (figure 1) and to assess the context within which the ABI is delivered.

**Ethics and dissemination** The APPRAISE protocol has been approved by the East of Scotland Research Ethics Committee (19/ES/0068), National Offender Management System (2019-240), Health Board Research and Development (2019/0268), Scottish Prison Service research and ethics committee, and by the University of Edinburgh's internal ethics department. The findings will be disseminated via peer-reviewed journal publications, presentations at local, national and international conferences, infographics and shared with relevant stakeholders through meetings and events.

**Trial registration number** ISRCTN27417180.

## Strengths and limitations of this study

► This study provides a unique opportunity to evaluate the feasibility of conducting trial-based research with men on remand in prison, a frequently under-represented group in research.

► A significant strength is that it will assess the feasibility and acceptability of the alcohol brief intervention from multiple perspectives, including men on remand, prison-based intervention teams and additional stakeholder groups.

► It will provide an opportunity to identify the operational criteria necessary for a future definitive randomised controlled trial.

► The unique context provides challenges for research, and the prisons' day-to-day running will influence research activities.

## INTRODUCTION
### Background/Rationale

The prevalence of at-risk drinking, which is defined as drinking at levels that harm a person's health, is far higher among those within the criminal justice system (73%)[1–5] than the general population (35%).[6] In the United Kingdom (UK), between 51% and 83% of incarcerated people are classified as risky drinkers,[7] and for those on remand in prison, the prevalence is between 62% and 68%.[5] Furthermore, alcohol dependence among those incarcerated (43%) is 10 times higher than the general population.[5]

The impacts of risky drinking are significant, resulting in significant health, economic and social burden on individuals, families and society as a whole.[8] Addressing alcohol harm in prisons, at what can be considered a 'teachable moment', could reduce the risk of reoffending, reduce costs to society and help address health inequalities.[9]

There is currently robust evidence from systematic reviews and meta-analyses, indicating that alcohol brief interventions (ABIs) are effective in reducing alcohol consumption among at-risk drinkers in healthcare settings.[8 9] However, there is little evidence of the effectiveness of ABIs in reducing risky drinking among those incarcerated, particularly those on remand,[10 11] despite there being evidence in the UK that ABIs can reduce recidivism.[12]

In addition, ABIs have been shown to reduce recidivism within the probation setting.[13] Our systematic review of the efficacy of psychosocial alcohol interventions for incarcerated people[10] found that interventions within prison have the potential to impact alcohol use positively; however, because of small numbers and different outcome measures, it was not possible to conduct a meta-analysis or generalise findings. Notably, none of the studies focused on men on remand, with remand referring to individuals who are unconvicted or convicted and unsentenced, held in custody awaiting trial and/or sentencing. The intervention in this proposed study builds on previous research exploring the theoretical validity of a self-efficacy enhancing ABI.[14 15] This intervention was initially tested within a pilot cluster randomised controlled trial (RCT), in a general hospital setting, and provided evidence of a potential effect.[15]

In the recently completed Alcohol Brief Interventions (ABIs) for male remand prisoners: an Medical Research Council (MRC) complex intervention framework development and feasibility study (PRISM-A), an ABI was developed and refined within the prison setting, working with men on remand to include a synthesis of their views, with reviews of the evidence base and theoretical underpinnings.[16 17] The findings also showed a high prevalence of at-risk drinking, with 82% of participants scoring ≥8 on the Alcohol Use Disorders Test (AUDIT).[17] This is comparable with other findings in the prison system in the UK[1–5] but is more than three times higher than primary care settings.[6] The study results indicated high levels of willingness of both staff and participants to engage with the intervention. Also, support was shown for an extended intervention, which helped men on remand to develop skills and strategies that would be useful for liberation. The feedback regarding frequency and intensity of contact identified a preference for more than one session, including additional community-based sessions, to allow participants to put their skills into practice when alcohol is more widely available. Based on learning from interviews within the PRISM-A study, the intervention for the APPRAISE pilot study includes three post-release booster sessions.

However, there is limited evidence around the optimum timing of delivery, recommended length, content, implementation and economic benefit of an ABI in the prison setting. Furthermore, there are weaknesses within the current evidence base, with interventions not having theoretical underpinnings.[11 18] This risks an intervention being used with a weak theoretical base, poorly specified 'active' ingredients and less likely to deliver the desired outcomes.

Following on from the positive results from the PRISM-A study, the MRC framework suggests the next step is to conduct a pilot study.[19] Therefore, the APPRAISE pilot trial has been proposed to strengthen the evidence around ABIs targeted at male remand prisoners. It will seek to assess the feasibility and acceptability of the ABI, in comparison to usual care, and it will explore the potential effectiveness of the key parameters. It will also aim to provide the information required to be able to design a definitive RCT.

### Aim

The APPRAISE study aims to undertake a two-arm, parallel-group, individually randomised, pilot study of a self-efficacy enhancing psychosocial alcohol intervention, for men on remand, in prison. If this pilot study is successful, it will provide the evidence to support the design of a future multicentre RCT.

The specific study objectives are as follows.

### Objective 1: to pilot the study measures and evaluation methods to assess the feasibility of conducting a multicentre, pragmatic, parallel-group, RCT

1a. Is it feasible to conduct a future multicentre RCT of a self-efficacy enhancing psychosocial ABI for men on remand?

1b. Can we obtain reasonable estimates of the parameters necessary to inform the design and sample size calculation for an RCT, including SD of potential continuous primary outcomes and estimates of recruitment, retention and follow-up rates?

1c. Can we determine, based on rates of missing data, whether the questionnaires are appropriate for the target population?

1d. Can we collect economic data needed for an RCT?

1e. Can we access recidivism (as measured by 'proven' reconviction rates[20] data derived from the Police National Computer (PNC) for participants)?

1f. Can we access health data from National Health Service (NHS) data sources for participants?

### Objective 2: to assess intervention fidelity

2a. What proportion of the interventions are delivered as per protocol?

2b. Is there any evidence of contamination between the two arms and/or between workers delivering the intervention?

2c. To what extent does the intervention change process variables consistent with the underpinning theory?

Objective 3: to qualitatively explore the feasibility and acceptability of the intervention and study measures to staff and participants on remand and on liberation

3. How acceptable are the trial and intervention procedures (including context and any barriers and facilitators) to the key stakeholders: men on remand in prison and on liberation; prison staff (including healthcare staff); commissioners; policy makers and third sector partners?

Objective 4: to assess whether operational progression criteria for conducting a future definitive RCT are met across trial arms and study sites and if so, develop a protocol for an RCT.

*Operational progression criterion are based on previous research results*[3]

4a. Do the two prisons invited to the study agree to take part?

4b. Based on knowledge from previous data, do at least 90 eligible participants consent to take part and be randomised?

4c. Do at least 70% of participants allocated to the intervention condition go on to receive at least one intervention session?

4d. Do at least 60% of participants take part in the follow-up assessments at 12 months across trial arms and study sites?

## METHODS AND ANALYSIS
### Study design

The study aligns to the MRC framework[19] using mixed methods within two linked phases conducted across two sites.

Phase I will involve a two-arm, parallel-group, individually randomised pilot study (Objectives 1 and 4). The pilot evaluation will provide data on feasibility and an assessment on the likely impact of the ABI to inform the feasibility and design of a future RCT.

Phase II will comprise a process evaluation (Objectives 2 and 3), drawing on the MRC process evaluation framework to guide the planning and design of the evaluation.[21] The aim of the process evaluation is: to assess how the ABI is implemented; to undertake a preliminary exploration of the change mechanisms underpinning the intervention (figure 1), and assess the delivery context of the ABI.

### Participants and setting
#### Identification of study participants

A suitable process of participant identification was established and found to be successful in the PRISM-A study.[16 17] The prison induction team (consisting of

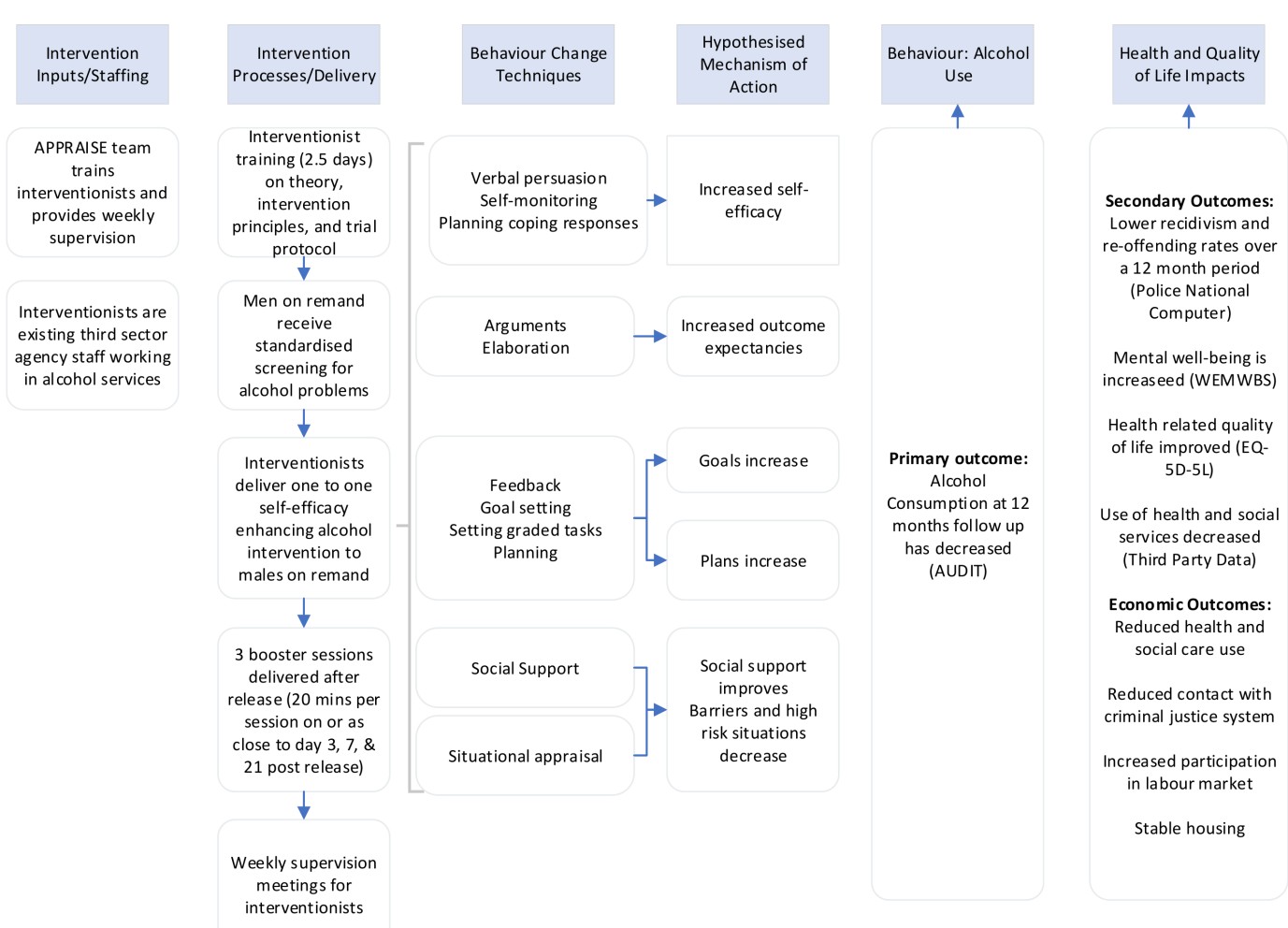

**Figure 1** APPRAISE logic model.

Intervention Inputs/Staffing
- APPRAISE team trains interventionists and provides weekly supervision
- Interventionists are existing third sector agency staff working in alcohol services

Intervention Processes/Delivery
- Interventionist training (2.5 days) on theory, intervention principles, and trial protocol
- Men on remand receive standardised screening for alcohol problems
- Interventionists deliver one to one self-efficacy enhancing alcohol intervention to males on remand
- 3 booster sessions delivered after release (20 mins per session on or as close to day 3, 7, & 21 post release)
- Weekly supervision meetings for interventionists

Behaviour Change Techniques
- Verbal persuasion, Self-monitoring, Planning coping responses
- Arguments Elaboration
- Feedback, Goal setting, Setting graded tasks, Planning
- Social Support
- Situational appraisal

Hypothesised Mechanism of Action
- Increased self-efficacy
- Increased outcome expectancies
- Goals increase
- Plans increase
- Social support improves, Barriers and high risk situations decrease

Behaviour: Alcohol Use
- **Primary outcome:** Alcohol Consumption at 12 months follow up has decreased (AUDIT)

Health and Quality of Life Impacts
- **Secondary Outcomes:** Lower recidivism and re-offending rates over a 12 month period (Police National Computer)
- Mental well-being is increased (WEMWBS)
- Health related quality of life improved (EQ-5D-5L)
- Use of health and social services decreased (Third Party Data)
- **Economic Outcomes:** Reduced health and social care use
- Reduced contact with criminal justice system
- Increased participation in labour market
- Stable housing

prison officers and peer prisoners) will provide potential participants with a verbal account of the study and a Participant Information Sheet (PIS) during their prison induction. Researchers will liaise with relevant prison officers each week to identify potential participants. Potential participants will be contacted via prison staff; those who are willing will meet with the researcher in an area of the prison where they can discuss issues in private, without risk of being overheard, and without risk to the interviewer or disruption to the running of the prison. The researcher will review the PIS with the participant, provide a written and verbal description of APPRAISE, answer any questions and invite him to consider participating. The researcher will then obtain written consent from willing participants. Those who do not wish to participate will receive no further interaction with the research staff. Data collection began on 1 December 2019 and will finish on 31 August 2021.

### Sample size

A total of 180 adult men (18 years and over) on remand will be recruited from two sites: one within the Scottish Prison Service (SPS) (n=90); and the second from Her Majesty's Prison and Probation Service (HMPS) in England (n=90). These sites were purposively selected following PRISM-A, as they offer geographical, socioeconomic and ethnic diversity, along with differing procedures and structures.[16] Both prisons provide alcohol services via external agencies who engage with people in custody and following release. Prisons have varying induction procedures, communication mechanisms and processes, and available, secure space to engage participants. Understanding the different approaches and dynamics will be important when designing a future RCT.

The target population is men on remand in prison, who have been in prison for 3 months or less. The average time on remand is approximately 9 weeks in England[22] and 4 weeks in Scotland.[23] We estimate that approximately 50% of participants will be released, while the rest will remain incarcerated; leaving 45 participants per study arm (90 in total) across the two sites.

This sample size was calculated to take account of the following factors[24]:

1. It will enable calculations of two-sided 95% CIs around proportions recruited, released and dropout in each study arm with half-widths of less than 0.15.
2. It exceeds the 30 per group recommendation of Lancaster *et al*[25] and 35 participants per group recommendation of Teare *et al*[26] for estimating key unknown design parameters, for example, SD with sufficient precision with a continuous primary outcome.
3. It will ensure that within each arm, across each site, we satisfy the minimum 12 per group 'rule of thumb' outlined by Julious[27] for pilot trials.

### Participant eligibility criteria

The following criteria were used in PRISM-A and were found to be appropriate.

### Inclusion criteria

Participants will be included in APPRAISE if they meet the following inclusion criteria:

► Informed consent given.
► Men aged 18 years or over.
► Have been in prison for 3 months or less on the current charge.
► Score 8 or more on the AUDIT.[28]
► Detained in either of the study sites within the SPS or HMPS.

### Exclusion criteria

Participants will be excluded from APPRAISE if they meet any of the following criteria:

► Previously recruited to APPRAISE.
► Unable to give informed consent or deemed incompetent/unable to make an informed decision regarding consent.
► Identified as a risk to self and/or others by prison staff.
► Judged to be under the influence of an illicit substance by prison or research staff.
► Currently taking disulfiram (frequently referred to as Antabuse).
► On a segregative rule (under prison rules).
► Not able to understand the documents (English language) or agree to the researcher aiding their understanding.

### Informed consent

All participants will be required to provide informed consent before taking part; their decision will be entirely voluntary and based on a clear understanding of what is involved. To ensure this, all participants will receive a verbal explanation of the study, in addition to a PIS. The verbal explanation will be given by the researcher and will cover all elements in the PIS and the consent form. Each willing participant will sign a consent form to ensure that informed consent has been obtained. Each participant will receive a copy of the form, and a copy will be filed in the Investigator Site File (ISF). All participants will be given sufficient time to consider the information provided, ask questions and clarify anything they do not understand. Participants will be reassured that they can withdraw their consent at any time, without loss of benefits to which they otherwise would be entitled.

### Ethics and dissemination

The APPRAISE protocol has been approved by the East of Scotland Research Ethics Committee (19/ES/0068), National Offender Management System (2019–240), Health Board Research and Development (2019/0268), Scottish Prison Service (SPS) research and ethics committee and by the University of Edinburgh's internal ethics department. The findings will be disseminated via peer-reviewed journal publications, presentations at local, national and international conferences, infographics and

**Table 1** Outline of APPRAISE intervention

| Element | Elements of intervention[14] | Enhancing self-efficacy | Delivery method and location |
|---|---|---|---|
| 1 | Preliminary discussion | Verbal persuasion | Face to face (P) Mobile phone (L) |
| 2 | Acquiring and providing information | Verbal persuasion | Face to face (P) |
| 3 | Self-monitoring | Verbal persuasion | Face to face (P) |
| 4 | Increasing awareness | Physiological state | Face to face (P) |
| 5 | Situation-appraisal and appropriate coping strategies* | Vicarious experience | Face to face (P) Mobile phone (L) |
| 6 | Goal setting* | Verbal persuasion | Face to face (P) Mobile phone (L) |
| 7 | Relapse* | Performance attainment | Face to face (P) Mobile phone (L) |
| 8 | Self-evaluation/ self-reinforcement* | Performance attainment | Face to face (P) Mobile phone (L) |
| 9 | Culmination | Performance attainment | Face to face and mobile phone (L) |

*Elements 5–8 highly rated by participants in the feasibility study (PRISM-A) and will form key focus of intervention delivery by mobile phone on liberation.
L, liberation; P, prison.

shared with relevant stakeholders through meetings and events.

## The APPRAISE intervention

The ABI has four sessions to be offered to all participants in the intervention arm. It will be delivered by existing staff at the voluntary sector organisation (VSO) who are currently engaged in alcohol service delivery in the prison and community setting. The ABI comprises nine elements (see tables 1 and 2) and will be delivered in four steps: Step 1 is 1×40 min, face-to-face session, covering nine elements, delivered by a trained (VSO) staff member within the prison. Steps 2, 3 and 4 are 20-minute sessions, by phone, on or as close to day 3, 7 and 21 post-liberation. Exact days will be recorded to inform the process evaluation. The post-liberation sessions will include elements 1 (preliminary discussion), 5 (situation-appraisal), 6 (goal setting), 7 (relapse), 8 (self-evaluation/self-reinforcement) and 9 (culmination).

Self-efficacy derives from Social Cognitive Theory and has been identified as an important determinant of health behaviour and health behaviour change.[29] The four primary sources of self-efficacy information (that can be targeted through interventions) are mastery experience, vicarious experience, verbal persuasion and physiological state.[14 15 29] The ABI is designed to increase self-efficacy—through the development of self-regulatory skills, self-management and self-belief, thus enabling an individual to address their alcohol consumption. Liberation then offers participants the ability to develop and build on these skills and self-belief through success and mastery, with their efforts leading to the adoption and maintenance of reduced alcohol consumption.[14 29] Reducing alcohol consumption can provide a sense of achievement and success, with the overall effects of such an intervention a likely increase in the men's level of self-efficacy.[14]

## Staff training

Two members of VSO staff at each site will be trained to deliver the ABI to participants (intervention), and the remaining staff will have no training (control). This method seeks to minimise contamination and has been used in other similar studies.[30 31] The possibility of residual contamination and the practicality of allocating staff according to randomisation will be explored within the process evaluation to inform the future RCT. Weekly debriefs with interventionists delivering the APPRAISE ABI will also enable the exploration of possible contamination.

## Intervention delivery

The ABI has four sessions to be offered to all participants in the intervention arm. Understanding what has been delivered and how is a key component of understanding the ABI implementation.[32 33] As part of the process evaluation to assess implementation, measures of fidelity (quality) and dose (quantity) will be recorded and analysed. To capture fidelity, we will digitally audio record 20% of intervention sessions and code these, using the Behaviour Change Counselling Index (BECCI),[34] to assess the extent to which the essential theoretical elements of the ABI were delivered as intended. To capture dose, quantitative data from participant study records and logs will be obtained to assess the proportion of interventions offered and delivered successfully. The number of intervention sessions offered, delivered and the length of each session will be recorded as will reasons for any unsuccessful delivery. Interviews with all implementers and 32 purposively selected participants will also be conducted at each site to identify how acceptable the trial intervention and procedures were.

## Control condition (care as usual)

The control condition will include participants receiving care as they usually would, within the existing service provision. What constitutes care as usual will vary between participants as some participants may access no support. In contrast, others may attend individual or group-based support sessions with VSO and/or other providers while in prison or their local communities. It should be noted that participants in the intervention condition will also have access to care as usual services. Service utilisation will be recorded for all participants.

| Table 2 | APPRAISE intervention elements |
|---|---|
| Element 1: Preliminary discussion | Opening strategies |
| Introduction to APPRAISE study | |
| Introduction to APPRAISE intervention | |
| Consent, confidentiality, engagement rules, trust | |
| Element 2: Acquiring and providing information | Feedback on AUDIT score |
| Establish perception of impact of alcohol on health and life | |
| Standard units of alcohol | |
| Recommended drinking levels | |
| Alcohol-related health problems | |
| Legal drink/drive limit | |
| Tips on reducing consumption | |
| Where to obtain information/support (prison and liberation) | |
| Element 3: Self-monitoring | Diary card—when, where, whom, type of drink, why |
| Element 4: Increasing awareness | Balance sheet—pros and cons of drinking |
| Physiological sensations identified | |
| Alternative appraisal of somatic sensations identified | |
| Strategies to reduce | |
| Element 5: Situation-appraisal and appropriate coping strategies | High-risk situations and antecedents of over-drinking identified |
| Alternative coping strategies identified | |
| Coping strategies verbalised by participant | |
| Praise provided | |
| Strategies developed further through coproduction | |
| Strategies modelled by interventionist | |
| Participant verbalises strategies and visualises them | |
| Plan for exposure/avoidance to low-risk situations and high-risk situations | |
| General control strategies: reduction in rate of drinking, sipping, low-alcohol content and alternating between soft or low-alcohol drinks | |
| Element 6: Goal setting | Setting realistic subgoals (short term) |
| Facilitating success and increasing motivation | |
| Element 7: Relapse | What happens if you relapse |
| What caused the relapse? | |

Continued

| Table 2 | Continued |
|---|---|
| How do I understand relapse? | |
| Element 8: Self-evaluation and self-reinforcement | Using my alcohol diary as a means of self-evaluation and self-reinforcement |
| Self-congratulations and rewarding my success | |
| What do I attribute my success to? | |
| Element 9: Culmination | Reflections and conclusions |
| Plans and goals reiterated and confirmed | |

AUDIT, Alcohol Use Disorders Test.

## Data collection
### Baseline assessment
Following appropriate consent procedures, baseline (TP0) data will be collected via researcher-led completion of questionnaires, and this will take approximately 30 min. Immediately after, participants will be randomised to the control or intervention arm and informed of their allocation. Participants randomised to the intervention arm will be provided with a face-to-face appointment to meet with their interventionist. This is the process used by the current providers ((VSO)when engaging with individuals in the prison setting. The three telephone sessions will be delivered following release from prison by one of the trained interventionists. In addition to the outcome measures (described below), data on age, gender, marital status, educational status and contact details will be collected.

### Blinding
Due to restrictions regarding electronic equipment and mobile phone use by researchers while in prison, a randomisation system, such as an interactive voice response telephone system or one accessed over the web, cannot be used. Allocation will be conducted at participant level using random permuted blocks of variable size[3 5 6] stratified by site using sealed envelopes, based on a predetermined random number allocation carried out by Edinburgh Clinical Trials Unit (ECTU).[27 35]

Researchers will not be involved in delivering the active and control interventions but will be aware of the study allocation of participants as the trial progresses. Allocation concealment will be used whereby neither the person delivering the interventions nor the participant will be aware of the study allocation until they are irrevocably entered into the trial. Both the trial statistician (RP) and health economist (JB) will be blinded to group allocation and will only have details of study participants by study number.

### Follow-up
Follow-up assessments will be conducted at 6 (TP1) and 12 months (TP2) post recruitment. The original aim was

for these to be conducted by the researcher either in person or by telephone. In light of COVID-19 restrictions, follow-up assessments have also been adapted to enable self-completion using hard copies or electronic versions. Where participants are identified as being incarcerated at follow-up, arrangements with the relevant prison will be made to undertake the follow-up. Attempts to contact participants will be multifaceted, including telephone calls, text messages and emails, based on participants' preferred method of contact, at varying times and days of the week. We will record the number of participants invited to participate; responded; were eligible; recruited; transferred; subsequently sentenced; liberated, and lost to follow-up. The number who returned to prison during the 12 months and who completed the trial will also be recorded.

## Outcomes

### Primary outcome measure

The proposed primary outcome measure is alcohol consumed, in units where one unit equals 8 g or 10 mL of ethanol, per week will be derived from the frequency and quantity of alcohol consumption questions in the extended version of AUDIT using a formula developed and employed in a number of RCTs of interventions for alcohol consumption.[36 37] A score of 8+ is referred to as a 'positive screen' and indicates drinking at hazardous (score of 8–15), harmful (score of 16–19) or probable dependent level (score of 20+). A score of 8 or more out of a possible 40 on the AUDIT has an established sensitivity of 92% and specificity of 94%[28] to detect those at risk of harm from alcohol use. The AUDIT will be administered at TP0, TP1 and TP2.

### Self-report secondary outcomes measures

Secondary outcome measures will be completed at TP0, TP1 and TP2.

The Warwick-Edinburgh Mental Well-being scale (WEMWBS) will be used to assess mental well-being.[38] This tool uses a 5-point Likert scale, which gives a score of 1 to 5 per question, giving a minimum score of 14 and maximum score of 70. A higher WEMWBS score indicates a higher level of mental well-being.[38]

Readiness to change will be measured using the Readiness to Change Ruler, which measures readiness to change drinking behaviour.[7]

Self-reported alcohol self-efficacy will be measured using the Drinking Refusal Self-Efficacy Questionnaire—Revised (DRSEQ-R)[39] and alcohol expectancy, using the Negative Alcohol Expectancy Questionnaire (NAEQ).[40]

The EuroQol five-dimensional five-level (EQ-5D-5L) will be used to measure health-related quality of life and is being scoped for potential future use to generate quality-adjusted life years (QALY).[41] A bespoke public sector service use questionnaire will be adapted based on the Economic Form 90[42] to determine public sector costs in the domains of health and social care, criminal justice system, unemployment and welfare.

### Other secondary outcome measures

Attempts will be made to access data from the PNC to collect individual-level data on recidivism rates as measured by 'proven' reconviction rates[20] over 12 months, since TP0. The number of participants incarcerated at TP1 and TP2 will also be recorded.

Individual-level data relating to health and social care use will also be sought. If access is possible, data derived from Community Health Index (CHI)/NHS number at TP3 will be used to validate the bespoke public sector service use questionnaire. Furthermore, data will be collected on the time spent by researchers and practitioners on APPRAISE to inform the resources needed for a definitive RCT.

### Mechanisms of impact

An intervention logic model (figure 1) informed by Social Cognitive Theory has been developed. The underlying causal mechanisms will be examined to provide increased understanding of how the intervention influences change through the quantitative assessment of key behavioural markers (mediators) of change, that is, self-reported self-efficacy, using the DRSEQ-R[39] and alcohol expectancy, using the NAEQ.[40] These behavioural markers of change will be recorded at TP0, TP1 and TP2. An exploratory prognostic analysis will be undertaken to explore the nature of change within these domains and their relationship to the primary outcome. The relationship between alcohol consumption, receipt of the intervention and reoffending will also be assessed. In addition, we will be interviewing 32 men who took part in the study (16 at each site), 8 intervention staff from CGL and 10 additional stakeholders. The qualitative exploration of participant responses to and interactions with the intervention will be undertaken to identify unanticipated pathways and consequences at 12 months of follow-up.

### Data analysis

The progress of participants through the APPRAISE pilot trial will be presented in accordance with the CONSORT (Consolidated Standards of Reporting Trials) guidelines[43] to allow descriptions of key parameters required for a future RCT; eligibility rates, consent, adherence, retention at follow-up and data completeness of outcome measures.

The statistical analyses will be primarily descriptive, providing an estimate of the proportion of those eligible, consenting, adhering to the intervention and retention rates at 6 and 12 months. These parameters will inform the power calculations for a future RCT and confirm or identify the need for modifications in relation to other aspects of the trial design, particularly the acceptability of study processes and outcome measures to participants and staff. Data about the flow of participants through the APPRAISE pilot trial will include numbers screened, the prevalence of the target condition, numbers providing contact details, numbers eligible and willing to consent and numbers followed up successfully at TP1 and TP2. In addition, data completeness

for each of the instruments and any potential bias in the follow-up data will be ascertained.

Analysis of the primary outcome will also provide estimates of the variance within and between groups for exploration of sample size calculations for a definitive RCT. Furthermore, we will calculate the variance of all primary and secondary outcome measures within groups to explore their utility in a definitive study. If access to PNC data is possible, proven reconviction rates will be measured by the number of offences committed by each treated participant against the number committed in the comparison group. Time-to-event analysis will also explore differences in offending patterns.

Calculation of between trial-arm differences (with 95% CIs) will indicate the likely effect sizes that will be observed in a future definitive trial.

For the primary outcome analysis, the mean difference (with 95% CI) in alcohol units consumed between the two randomisation groups will be calculated, both overall and stratified by site. A complete case analysis will be used, but a sensitivity analysis will also be incorporated to explore the effects of missing data. This will involve using multiple imputations for the primary outcome in a supplementary analysis, provided that the valid sample size is sufficient to support this. In addition, we will add and subtract one-unit difference (alcohol units) to the missing data as part of a sensitivity analysis to explore how the 95% CIs change depending on our assumptions about the missing data (bearing in mind that the imputed data cannot be less than zero). This is similar to the sensitivity analysis method suggested in White *et al* (2011) which involves determining how large an amount can '*be added to or subtracted from the imputed data without changing the clinical interpretation of the trial*'. We will adopt a similar approach to determine how sensitive the conclusions of the APPRAISE pilot trial (and the decision about whether to proceed with an RCT) are to our assumptions about missing data.

A statistical analysis plan will be written and finalised prior to analysis.

### Health economic evaluation

Full details of all health economic analyses will be specified in a health economic analysis plan, authored by the trial health economist(s), and signed off by the Principal Investigator prior to analysis.

In brief, this will scope the feasibility of conducting a future health economic evaluation alongside any future trial. A bespoke survey of public sector service use will be adapted from the Economic Form 90[42] and tested in the patient population. We will also scope the potential for using the EQ-5D-5L relative to other health outcome measures being used in the trial. The rates of response and completeness of questionnaires will be reported in each case. A dry run costing/QALY generation exercise will be undertaken to identify any practical issues in using these instruments. The dry run costing exercise will include a detailed scoping of the cost elements relating directly to the introduction and running of the intervention through consultation with key staff involved in the implementation. Finally, the potential for longer-term

modelling of outcomes will be scoped through consultation with experts and non-systematic searches of the literature for similar work or anticipated parameters. These will form the basis for a short report to aid in developing a future protocol for a follow-up trial.

### Additional analyses

All interview data will be transcribed and thematic analysis undertaken in order to fulfil Objectives 1d, 2c and 3a. Braun and Clarke's thematic analysis framework will guide this, as it is congruent with the study's theoretical perspective.[44] To facilitate an understanding of each stakeholder group's perspective, the data will initially be analysed within group. The rationale for undertaking the analysis within each group is to enable the targeting of remedial actions within a future RCT. Themes running across the data, in relation to each question, will then be collated, providing a multiperspective understanding of the stakeholder experience for each element of the intervention/study.

The qualitative analysis will provide evidence to verify the feasibility and acceptability of the ABI to participants, specifically elements 5–8. It will also inform the process evaluation in phase II to better understand and explain the social processes of the intervention, through the lens of implementation theory, Normalisation Process Theory.[45–48]

### Monitoring

An APPRAISE Trial Steering Committee (TSC) has been set up to monitor the implementation, to provide an independent assessment of the data analysis and to determine if a future trial is merited.

The TSC has the following objectives:
- ► Provide supervision of the trial on behalf of the sponsor and funder and ensure it is conducted to rigorous standards.
- ► Monitor the ongoing progress and adherence to protocol.
- ► Consider new information of relevance to the research question.
- ► Provide advice, through the chair, to the chief investigator and funder on all appropriate aspects.
- ► Provide evidence to support extension requests.

In addition to the TSC, the chair plus three TSC members (specialising in health economics, statistics, and qualitative methodology and analysis) have formed a subcommittee responsible for data monitoring and ethics (DMEC).

The DMEC subcommittee has the following objectives:
- ► To monitor the data and recommend the TSC on whether there are any ethical or safety reasons why the trial should not continue.
- ► To ensure that the safety, rights and well-being of participants remain paramount.

The subcommittee will also consider the need for any interim analysis and are the only body who, if necessary, will have access to the unblinded data.

## Data management

Data from both study sites will be anonymised, and individuals' data will only be identifiable by their unique screening number. All of the hard copy data from the England site will be sent to the coordinating site, The University of Edinburgh, by secure courier, where all of the data will then be stored in a locked filing cabinet, with restricted access. Personal or sensitive data will be transported separately between sites securely, using password-protected files. No personal data will be transferred outside the UK, or stored or collected on computer servers outside the UK.

Audio files will be uploaded securely to a password-protected site by the researchers, where the approved transcription company will be able to access the files. Transcribed files will be returned to the researchers using the same secure platform and downloaded and stored securely in folders on the University of Edinburgh's password-protected server. Following a quality check to ensure the transcribed files' accuracy, all audio recordings will be securely destroyed.

## Confidentiality and data protection

All study staff will comply with the requirements of the appropriate data protection legislation (including the General Data Protection Regulation and Data Protection Act) concerning the collection, storage, processing and disclosure of personal information. All evaluation forms, reports and other records will maintain participant confidentiality and only be identifiable via a unique participant identification number. All written records will be kept in a secure storage, and computers used to collate the data will be password protected. Published results will not contain any personal data to ensure that individuals cannot be identified. Personal data will be stored for 24 months after the study completion, and following this will be securely destroyed.

## Ethical considerations

Ethical approval has been obtained from the East of Scotland NHS Research Ethics Committee, the Scottish Prisons Service Research Ethics Committee, Her Majesty's Prison and Probation Service and the University of Edinburgh School of Health in Social Science Research Ethics Committee.

Due to the high-risk population group, study staff will document any occasions where a session has had to be stopped due to participant distress or where the session is terminated because the researcher or prison staff identify a risk to the researcher. This information will be communicated to the chair of the TSC within a week of it occurring. In Scotland, it will be possible to identify and document whether participants are put on a suicide risk management strategy while they are in prison. This will be monitored and any participant put onto this strategy within a week of being in contact with study staff will be flagged and reported to the chair within 24 hours.

## Dissemination

Our dissemination plan will include local, national and international avenues. The findings will be prepared for publication via open access, peer-reviewed journal articles. Also, the APPRAISE pilot trial findings and any plans for a future RCT will be presented to Health and Justice Teams at Scottish Government and Public Health England, the National Prisoner Healthcare Network (Scotland) and the WHO (Health in Prisons Programme Collaborating Centre). We will ensure participants can engage with research findings through infographics and will feedback to the participating sites. Other dissemination opportunities include presentations at meetings, workshops, national and international conferences, or at relevant organisations and events, such as Scottish Alcohol Research Network, Scottish Health Action on Alcohol Problems and Offender Health Research Network.

## Patient and public involvement

Patient and public involvement has informed and influenced the development of the study. Representatives have included those with lived experience of imprisonment and those involved in the delivery of alcohol support and advice to criminal justice populations. Their expertise has informed the development of study materials and will also inform the analysis, reporting and dissemination of the research.

**Author affiliations**
[1]School of Health in Social Science, University of Edinburgh, Edinburgh, UK
[2]School of Social Sciences, Humanities & Law, Teesside University, Middlesbrough, UK
[3]Institute for Gender Medicine, Charite Universitatsmedizin Berlin, Berlin, Germany
[4]Centre for Change, County Durham Drug and Alcohol Recovery Service, Durham, UK
[5]Division of Community Health Sciences, University of Edinburgh, Edinburgh, UK
[6]Edinburgh Clinical Trials Unit, University of Edinburgh, Edinburgh, UK
[7]Edinburgh Health Services Research Unit, The University Of Edinburgh, Edinburgh, UK
[8]Public Health Directorate, NHS Lothian, Edinburgh, UK
[9]Centre for Health Service Studies, University of Kent, Canterbury, UK
[10]University of Aberdeen, Aberdeen, UK
[11]Institute of Social Marketing, University of Stirling, Stirling, UK
[12]Department of Economics, University of North Carolina at Greensboro, Greensborough, UK
[13]Therapeutic Solutions, London, UK
[14]Criminal Justice, Alcohol, Drugs and Tobacco Division, Public Health England, London, UK

**Contributors** AH managed the overall design and inception of the study with DN-B. AH and DN-B were responsible for coordinating the Scottish and English study sites, respectively. VG, GW, and JB were the study coordinators (VG—Scotland; GW and JB—England). RAP and JR are the trial statisticians. JB and AStoddart are the health economists. ASheikh, KH, PS, RS and SC have provided methodological expertise. PC, ASondhi and KL have informed third-party data collection. JB, GS, JF and SM have supported the development of intervention and participant materials. All coauthors also contributed to the revisions and final draft of the paper.

**Funding** This study is funded by the NIHR Public Health Research Programme (17/44/11).

**Competing interests** None declared.

**Patient and public involvement** Patients and/or the public were involved in the design, or conduct, or reporting or dissemination plans of this research. Refer to the Methods section for further details.

**Patient consent for publication** Not required.

**Provenance and peer review** Not commissioned; externally peer reviewed.

**ORCID iDs**
Victoria Guthrie http://orcid.org/0000-0002-3920-2342
Rosie Stenhouse http://orcid.org/0000-0002-1253-2544

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
