## [Reviewer comments · BMJ Open]

ARTICLE DETAILS

TITLE (PROVISIONAL)	A two-arm parallel group individually randomised Prison Pilot study of a male Remand Alcohol Intervention for Self-efficacy Enhancement: The APPRAISE Study Protocol
AUTHORS	Holloway, Aisha; Guthrie, Victoria; Waller, Gillian; Smith, Jamie; Boyd, Joanne; Mercado, Sharon; Smith, Pam; Stenhouse, Rosie; Sheikh, Aziz; Parker, Richard; Stoddart, Andrew; Conaglen, Philip; Coulton, Simon; Stadler, Gertraud; Hunt, Kate; Bray, Jeremy; Ferguson, Jennifer; Sondhi, Arun; Lynch, Kieran; Rees, Jessica; Newbury-Birch, Dorothy

VERSION 1 – REVIEW

REVIEWER	Lorna Hardy University of Exeter, UK
REVIEW RETURNED	30-Jun-2020

GENERAL COMMENTS	The purpose of this study is to determine the feasibility and acceptability of an alcohol-focused brief intervention for male prisoners on remand. There is currently little evidence on the effectiveness of such interventions in prisoners, and thus this research addresses a significant gap in the literature. This study is well designed to address the research questions posed, and the protocol well-written and structured. I have some minor comments. 1. For objective 1c (pg 7), can the authors clarify how completing questionnaires 'well' will be operationalised in the present study? This is broken down in greater detail in the data analysis section, but it would be good to include it here briefly as well (e.g. if this judgement will be based on rates of missing data). I would also consider reframing this point to avoid value judgements of the population (i.e. are the questionnaires appropriate for the target population rather than whether they complete them 'well'). 2. In the section on exclusion criteria (pg 16) please could the authors specify on what basis potential participants will be identified as a risk to self and/or others. Will this be based on pre-existing assessments undertaken by staff, for example? It is important to specify this I think to remove any concern of unconscious bias in recruitment (i.e. exclusion of 'difficult' participants who staff may consider less likely to benefit from the active intervention). 3. In the self-report secondary outcome measures section (pg 18) I am interested as to why the researchers have chosen to include only a negative expectancies of alcohol questionnaire (the NAEQ), and not an equivalent questionnaire of positive expectancies. Is it expected that the intervention will specifically increase negative
---

	expectancies of alcohol (rather than also reducing positive expectancies)? It would be great if the authors could clarify this. 4. In the section on other secondary outcome measures (pg 19) – is it possible to specify which health data the researchers will be collecting (and how this data relates to the research questions)? Is it use of alcohol treatment services or something else?
--	--

REVIEWER	Claire Keen The University of Melbourne, Australia
REVIEW RETURNED	30-Aug-2020

GENERAL COMMENTS	This paper outlines the protocol for a pilot trial of an alcohol brief intervention for men on remand. There are the limited programs offered to people on remand and the high levels of alcohol dependence and risky-alcohol use. This study is an important step in developing appropriate and effective alcohol programs for people on remand. Some specific comments are listed below. Introduction:  - Expand ABI the first time it is used in the main text. - The information presented in paragraphs 2-5 of the introduction could be re-ordered to make it clearer what relates specifically to ABI research and what is broader research on alcohol interventions in prison. For example, the information on the systematic review of psychosocial alcohol interventions for incarcerated people discussed in paragraph 3 and the first sentence of paragraph 4 may fit better before paragraph 2, as it discusses the lack of evidence on alcohol interventions in prison more broadly. Whereas paragraph 2 and the second sentence of paragraph 4 are specific to ABI. - In paragraph 2 the authors note the “weaknesses within the current evidence base, with regards to the theoretical underpinnings”. Can the authors please explain in more detail what they mean by this. Do they mean that the current evidence base is not driven by theory, or that the theory is incorrect or some combination of these factors? - The last sentence of paragraph 2 that introduces the APPRAISE pilot trial could be moved towards the end of the introduction, as it is confusing to introduce the current trial and then return to a literature review. - Paragraph 5: the authors note that PRISM-A recruited people in prison who scored >8 on the AUDIT, then note that the same study found that 82% of participants scored >=8. Is this referring to participants screened for eligibility? If so, some discussion of potential limitations of this estimated prevalence may be necessary. Presumably those screened for eligibility were likely to have risky drinking behaviour (as they were deemed potentially eligible for the study). - Similarly, the last sentence of paragraph 5 of the introduction could fit better in the methods, as it describes elements of the APPRAISE design, rather than the introduction. Methods and analysis  - The description of the APPRAISE intervention may fit better within the Methods and analysis section. As it is, the sections following the
--

	aims jump around from a detailed description of the specific intervention, to a high level description of the actual study design.  - Consider organising the Methods and Analysis section using a “PICO” structure. This would aid the reader to find the information they wanted quickly and to know what was coming next. Information related to ethics and confidentiality may fit better with the monitoring and data management sections. - Sample size: it would be useful to have the discussion on the total sample size before the discussion of the expected proportion released, as the way that sentence is written presumes a knowledge of the total sample size. - The sections on baseline and follow-up assessments may fit better in or near the outcome section. - Consider including a diagram that shows the timing of different baseline and outcome assessments. (e.g. which assessments would occur at Baseline, TP1 and TP2 and the timing of these assessments in relation to intervention completion). - Data management. The data management section only refers to data at the “second study site” How will data management be addressed at the first site? - Please include the planned dates of the study.
--	--

VERSION 1 – AUTHOR RESPONSE

Reviewer 1

Comment 1 - Thank you very much for this observation. The wording of the objective on P.7 has been changed to reflect this.

Comment 2 - Thank you very much. Clarification has been made on P.16

Comment 3: The reviewer is correct in highlighting our plan to collect data on only negative alcohol expectancies. The reasons are twofold. First the evidence of expectancies as predictors of changes in drinking behaviour is far stronger for negative rather than positive expectancies. Positive expectancy is associated in maintenance rather than change in drinking behaviour. Second, one of the key aims of an alcohol brief intervention is to create a sense of ambivalence in the participant, a dissonance between behaviour and beliefs that direct that behaviour, this is better achieved through the enhancement of negative expectancy than the diminishing of positive expectancy. We expect the intervention will increase negative alcohol expectancies and in combination with enhanced self-efficacy this will lead to reductions in alcohol consumption.

Comment 4: Clarification has been made on P.18.

Reviewer 2:

Comment 1: Thank you for highlighting this. This has been amended on P.7 (paragraph 2).

Comment 2- 6: We have rewritten the introduction taking these comments into account.

Comment 7: We have moved to the methods and analysis section, directly following the different

phases of the study and the population as per the request below for it to follow PICO.

Comment 8: We have organised the existing sections into the 'PICO' order as flagged by the comments in paper. Also moved the ethics and confidentiality into data management sections as requested.

Comment 9: Thank you, we have rectified this and it now reads:

“180 adult men (18 years and over) on remand will be recruited from two sites: one within the Scottish Prison Service (SPS); Edinburgh (n=90) and the second from Her Majesty’s Prison and Probation Service (HMPS); Durham, England (n=90). These sites were purposively selected following PRISM-A, as they offer geographical, socio-economic and ethnic diversity, along with differing procedures and structures [15]. Both prisons provide alcohol services via external agencies who engage with people in custody and following release. Prisons have varying induction procedures, communication mechanisms and processes, and available secure space to engage with participants. Understanding the different approaches and dynamics will be important when designing a future RCT.

The target population is men on remand in prison, who have been in prison for three months or less. The average time on remand is approximately nine weeks in England [28] and 4 weeks in Scotland [29]. We estimate that approximately 50% of participants will be released, while the rest will remain incarcerated; leaving 45 participants per study arm (90 in total) across the 2 sites.”

Comment 10: We have moved these sections to directly above the outcomes.

Comment 11: Thank you for this comment. We have taken the decision to exclude the TLFB is an increased need to use tools appropriate for self-completion as a result of COVID-19. Given the exclusion of the TLFB, all primary and secondary instruments will be collected at TP0, TP1, and TP2.

Comment 12: We have reworded to be clearer, the text now reads-

“Data from both study sites will be anonymised and individual’s data will only be identifiable by their unique screening number. All of the hard copy data from the Durham site will be sent to the coordinating site, The University of Edinburgh, by secure courier, where all of the data will then be stored in a locked filing cabinet, with restricted access. Personal or sensitive data will be transported separately between sites in a secure manner, using password protected files. No personal data will be transferred outside the UK, or stored or collected on computer servers outside the UK”.

Comment 13: This has now been included on P.23.

VERSION 2 – REVIEW

REVIEWER	Lorna Hardy University of Exeter, UK
REVIEW RETURNED	21-Dec-2020

GENERAL COMMENTS	I am satisfied that the authors have addressed my comments, and feel the manuscript is substantially improved in its current form.
--

REVIEWER	Claire Keen University of Melbourne, Australia
REVIEW RETURNED	27-Dec-2020

GENERAL COMMENTS	Thank you for addressing the reviewer comments. Please carefully
--

	proofread the manuscript, as there are some occasions where the grammar is not quite right or where the information in text does not exactly match that in the table (e.g. slight differences between Table 1 and the description in text). Apart from that I have no further comments.
--	---